# DNA deaminases induce break-associated mutation showers with implication of APOBEC3B and 3A in breast cancer kataegis

Benjamin JM Taylor[1†], Serena Nik-Zainal[2†], Yee Ling Wu[1], Lucy A Stebbings[2], Keiran Raine[2], Peter J Campbell[2], Cristina Rada[1], Michael R Stratton[2], Michael S Neuberger[1]*

[1]Protein and Nucleic Acid Chemistry Division, Medical Research Council Laboratory of Molecular Biology, Cambridge, United Kingdom; [2]Cancer Genome Project, Wellcome Trust Sanger Institute, Cambridge, United Kingdom

**Abstract** Breast cancer genomes have revealed a novel form of mutation showers (kataegis) in which multiple same-strand substitutions at C:G pairs spaced one to several hundred nucleotides apart are clustered over kilobase-sized regions, often associated with sites of DNA rearrangement. We show kataegis can result from AID/APOBEC-catalysed cytidine deamination in the vicinity of DNA breaks, likely through action on single-stranded DNA exposed during resection. Cancer-like kataegis can be recapitulated by expression of AID/APOBEC family deaminases in yeast where it largely depends on uracil excision, which generates an abasic site for strand breakage. Localized kataegis can also be nucleated by an I-SceI-induced break. Genome-wide patterns of APOBEC3-catalyzed deamination in yeast reveal APOBEC3B and 3A as the deaminases whose mutational signatures are most similar to those of breast cancer kataegic mutations. Together with expression and functional assays, the results implicate APOBEC3B/A in breast cancer hypermutation and give insight into the mechanism of kataegis.

*For correspondence: msn@ mrc-lmb.cam.ac.uk

†These authors contributed equally to this work

Competing interests: The authors declare that no competing interests exist.

## Introduction

Whole genome sequencing of 21 breast cancers recently revealed the presence in more than half the cancers of a novel form of localised hypermutation (termed kataegis) in which clusters of multiple same-strand mutations at C:G pairs are focused on multikilobase-long genomic regions with adjacent mutations within each cluster separated by one to several hundred base pairs (*Nik-Zainal et al., 2012*). Although the mechanism underlying kataegis is unknown, the fact that the mutations occurred nearly exclusively at C residues preceded by a 5′-T suggested that AID/APOBEC cytidine-DNA deaminases might possibly be involved in the process since these enzymes are sensitive to the 5′-flanking sequence context (*Conticello et al., 2007*; *Nik-Zainal et al., 2012*).

Members of the AID/APOBEC family of enzymes (reviewed in *Conticello et al., 2007*) deaminate cytosine in the context of a single-stranded polynucleotide substrate, and function in adaptive and innate immunity. AID acts on C residues in the DNA of the genomic immunoglobulin loci in activated lymphocytes to trigger antibody gene diversification whereas APOBEC3 family members, of which there are seven in humans, act on C residues in the DNA of viral replication intermediates (usually in the cytoplasm) as part of a host restriction pathway. Off-target deamination by AID results in nucleotide substitutions and genomic rearrangements in B lymphocyte tumours, some of which are implicated in oncogenesis (reviewed by *Gazumyan et al., 2012*). Although AID is the only member of the AID/APOBEC

**eLife digest** The genomes of cancer cells contain mutations that are not present in normal cells. Some of these prevent cells from repairing their DNA, while others give rise to tumours by causing cells to multiply uncontrollably. Moreover, some of the mutations in breast cancer cells occur in clusters—a phenomenon known as kataegis (from the Greek for 'thunderstorm').

Kataegic mutations occur almost exclusively at a cytosine preceded by a thymine. This suggests that a family of proteins called AID/APOBEC enzymes—which remove amine groups from cytosines—may be involved in generating these mutations. In this study, Taylor et al. confirm this possibility by showing that expressing individual members of the AID/APOBEC family of enzymes in yeast cells increases the mutation frequency and induces kataegis.

The kataegis triggered by the AID/APOBEC enzymes could be localised through the introduction of double-stranded breaks into the DNA: Taylor et al. suggest that this might happen because repairing the breaks exposes single-stranded DNA, which the AID/APOBEC enzymes then act upon. By comparing the mutations induced in the yeast cells with those observed in breast cancer cells, Taylor et al. identified APOBEC3B as the enzyme most likely to be responsible for kataegis in breast cancer (with APOBEC3A also a strong candidate in some cancers). Moreover, they showed that APOBEC3B was highly expressed in breast cancer cell lines, and that APOBEC3B and APOBEC3A can also cause DNA damage in human cells.

Taken together, the findings provide key insights into the mechanism by which kataegis arises, and identify two proteins likely to contribute to the mutations seen in breast cancer. Further work is now required to determine whether these enzymes also give rise to mutations in other forms of cancer.

family known to act physiologically on endogenous nuclear DNA, it is possible that other members of the AID/APOBEC family might occasionally get access to the nucleus and cause cancer-associated genomic damage or mutation (*Harris et al., 2002*; *Beale et al., 2004*; *Vartanian et al., 2008*; *Stenglein et al., 2010*; *Landry et al., 2011*; *Nik-Zainal et al., 2012*; *Nowarski et al., 2012*).

Here we have asked whether we could recapitulate cancer-like kataegis by expression of different AID/APOBEC enzymes in budding yeast and if so, use the tractability of yeast to gain insight into the kataegic process. The results provide insight not only into the mechanism of kataegis but also provide a strong pointer to the identity of the deaminases responsible for the kataegis observed in breast cancers.

## Results

### Recapitulating kataegis in yeast with AID/APOBEC deaminases

Several members of the AID/APOBEC family members were expressed in yeast and all were found to give a significant increase in the mutation frequency as judged by the yield of colonies resistant to canavanine (Can$^R$) (*Figure 1—figure supplement 1*). Genome sequencing, however, revealed that most Can$^R$ colonies had typically accumulated less than 10 point mutations (>98% at C:G pairs) during the period of AID/APOBEC induction and clonal expansion (*Figure 1A*). A hyperactive mutant of AID (AID*; *Wang et al., 2009*) gave a significantly higher mutation load (median of 25 mutations per genome). We therefore initially focused on the mutations in AID*-transformants (1078 mutations in total, of which all except 14 were at C:G pairs) to see if there were signs of kataegis.

The distances between neighbouring mutations in the AID* yeast transformants are displayed as rainfall plots (*Figure 1B*). While the median overall intermutational distance (IMD) is 13 kb, it is apparent that rather than the mutations being scattered randomly over the genome, mutation distribution is bimodal (*Figure 1C*). Dividing the mutations into two groups using k-means cluster analysis (*Hartigan and Wong, 1979*) reveals that one group exhibit a median IMD of 156 kb with a distribution of distances that is as expected for a set of individual mutations randomly scattered over the yeast genome as judged by Monte Carlo simulation (*Figure 1C*). We designate these as singlet mutations: they account for 52% of the total mutations. The remaining 48% of the mutations are much more closely spaced than would be expected on a random basis. We designate these as proximal mutations, which are separated from each other by a median IMD of only 727 bp with >99% of them being within 8.5 kb of their closest neighbour.

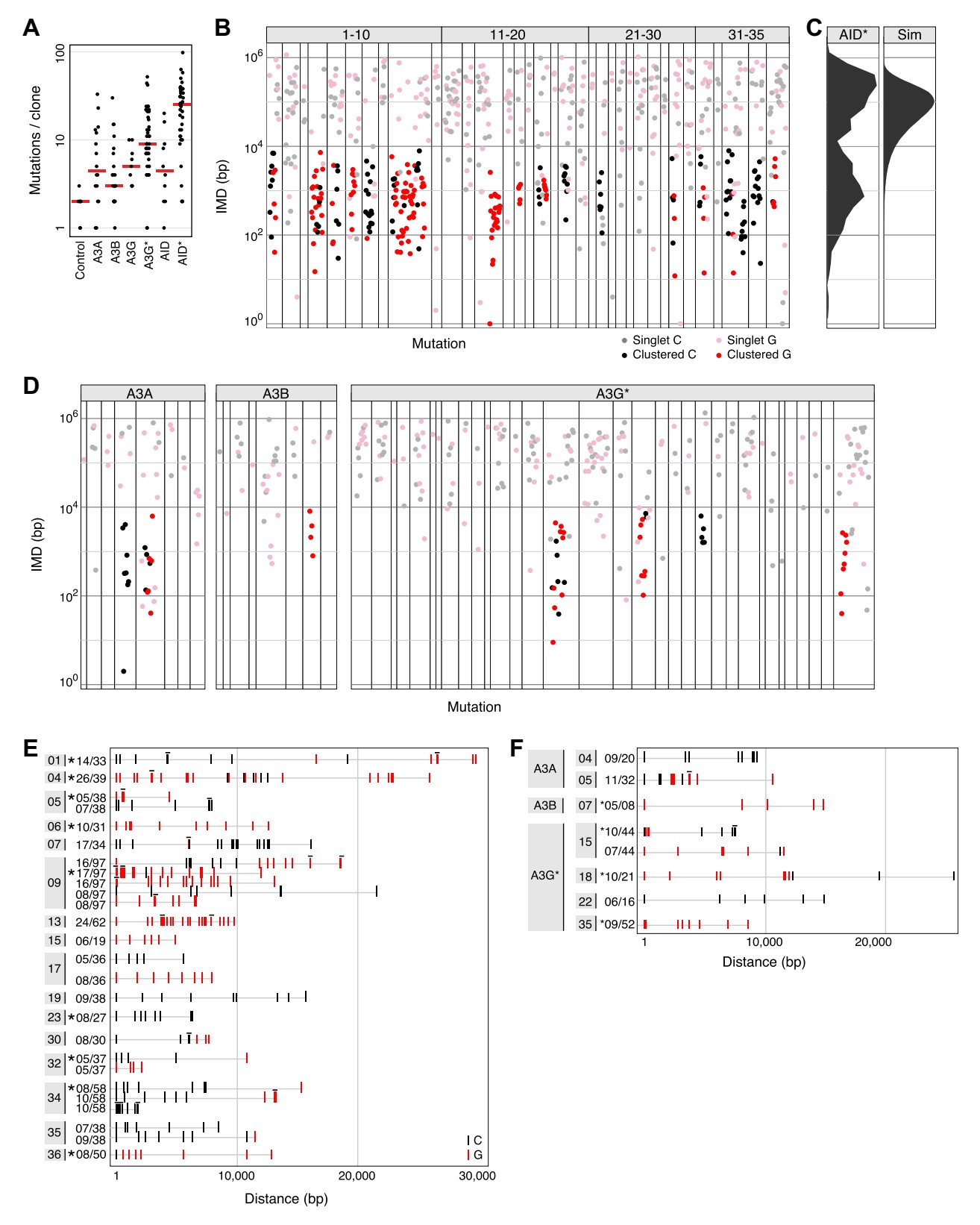

**Figure 1**. AID/APOBEC-induced kataegis in yeast. (**A**) The total number of mutations detected in canavanine-resistant (Can^R) AID/APOBEC yeast transformants, median frequency indicated. (**B**) Rainfall plot of genome-wide intermutational distances (IMD) in individual Can^R AID* transformants. *Figure 1. Continued on next page*

*Figure 1. Continued*

Clone identifier is indicated along the top, mutations shown as dots with the y-axis giving the distance to the next downstream mutation on the same chromosome. For each clone, dots are ordered sequentially along the genome. Dot colours represent: cluster mutations (at C, red; G, black), unclustered mutations (at C, pink; G, grey). Mutations at A:T (14 out of 1078 total), single mutations on individual chromosomes (15% of the database), the most downstream mutation in each chromosome and transformants without any multiply mutated chromosomes (5/40) are not depicted. *Supplementary file 1B* contains the location of all identified mutations. (**C**) Observed distribution of IMDs in the AID* dataset compared to a simulation assuming mutations are randomly scattered throughout the genome. (**D**) IMD plots of APOBEC3A/B/G*-expressing yeast transformants (28/78 transformants harboured no multiply mutated chromosome and are not depicted). (**E**) Detailed view of AID* mutation clusters. Each line represents an individual cluster with the clone identifier (grey box), number of mutations in the cluster and total mutations in the clone indicated. Mutations are coloured as in (**B**), a horizontal line indicates mutations that have coalesced, * indicates clusters localising within 10 kb of *CAN1*. All clusters containing ≥5 mutations are depicted. (**F**) Mutation clusters identified in yeast APOBEC3 transformants.

The following figure supplements are available for figure 1:

**Figure supplement 1**. Characterisation of AID/APOBEC yeast transformants.

The overwhelming majority of the proximal mutations in the AID* transformants do not occur as isolated mutational pairs but, rather, are found in clusters. Thus, if we define proximal mutations as a pair of mutations that are located <8.5 kb apart (a distance that excludes 99% of the singlet mutations) and define a cluster as a stretch of DNA containing ≥5 proximal mutations, we find that 75% of the AID*-induced proximal mutations are actually parts of clusters. These clusters typically extend over 6–15 kb (with the full range detected being 1.8–30 kb) and contain anything up to 26 mutations (*Figure 1E*). This clustering is far in excess of anything that would be expected on a random basis. The level of mutation clustering observed with AID* is such that more than one-third of the transformants analysed (16/40) contain at least one mutation cluster. In affected clones, a quarter to two-thirds of all the mutations in the cell are concentrated in a small number of clusters that account for <0.2% of the entire genome. Similar clusters were also observed in yeast cells transformed with APOBEC3A and APOBEC3B as well as with the hyperactive APOBEC3G mutant APOBEC3G* (*Figure 1D,F*). Like the cancer kataegis, the clustered mutations in the various yeast transformants showed a strong tendency towards strand polarity; mutations within a cluster occur predominantly at either a C residue or a G residue with over 88% of mutations being strand coordinated (*Figure 1D,F*).

## Transversion mutations are preferentially associated with kataegic stretches

Exploring the mutational spectra, we find that the majority (76%) of the mutations in the yeast AID* transformants are C→T transitions, although transversions do occur and these are preferentially associated with the kataegic stretches (*Figure 2A* and *Table 1*). Transversions account for 54% of the kataegic mutations in the AID* transformants but for only 13% of the unclustered substitutions (*Table 1*). The same bias towards transversion mutations in the kataegic stretches is also observed in the APOBEC3A, 3B and 3G* transformants (*Figure 2A* and *Table 1*).

## Transversion mutations are dependent on UNG and REV1

Whereas C→T transitions will likely arise through direct replication over uracils generated by cytidine deamination, transversions are presumably due to replication over abasic sites created through uracil excision by uracil-DNA glycosylase (UNG). The transversions exhibit a strong (4- to 10-fold) bias for C→G rather than C→A substitutions (*Table 1*) suggesting that the replication over the abasic site could be catalysed by REV1 since this translesion polymerase (by virtue of its deoxycytidyl nucleotide transferase activity) inserts C opposite abasic sites (*Nelson et al., 1996*). Indeed, deficiency in either REV1 or UNG led to a dramatic fall in the proportion of transversion mutations (*Table 1*). Deficiency in UNG also resulted in a fourfold increase in the average total mutation load in AID* transformants (*Supplementary file 1B*). This presumably reflects diminished repair of the AID/APOBEC-generated uracils. There was an overall decrease in average total mutation load in AID* transformants of REV1 deficient yeast that might reflect the possible non-catalytic roles of REV1 during DNA damage repair (*Sale et al., 2012*).

## UNG-Deficiency diminishes kataegis in yeast

Since UNG is required for the transversion mutations that are enriched in kataegic stretches, we asked whether UNG itself is required for kataegis. We found that the increased mutation load in AID* *ung1Δ*

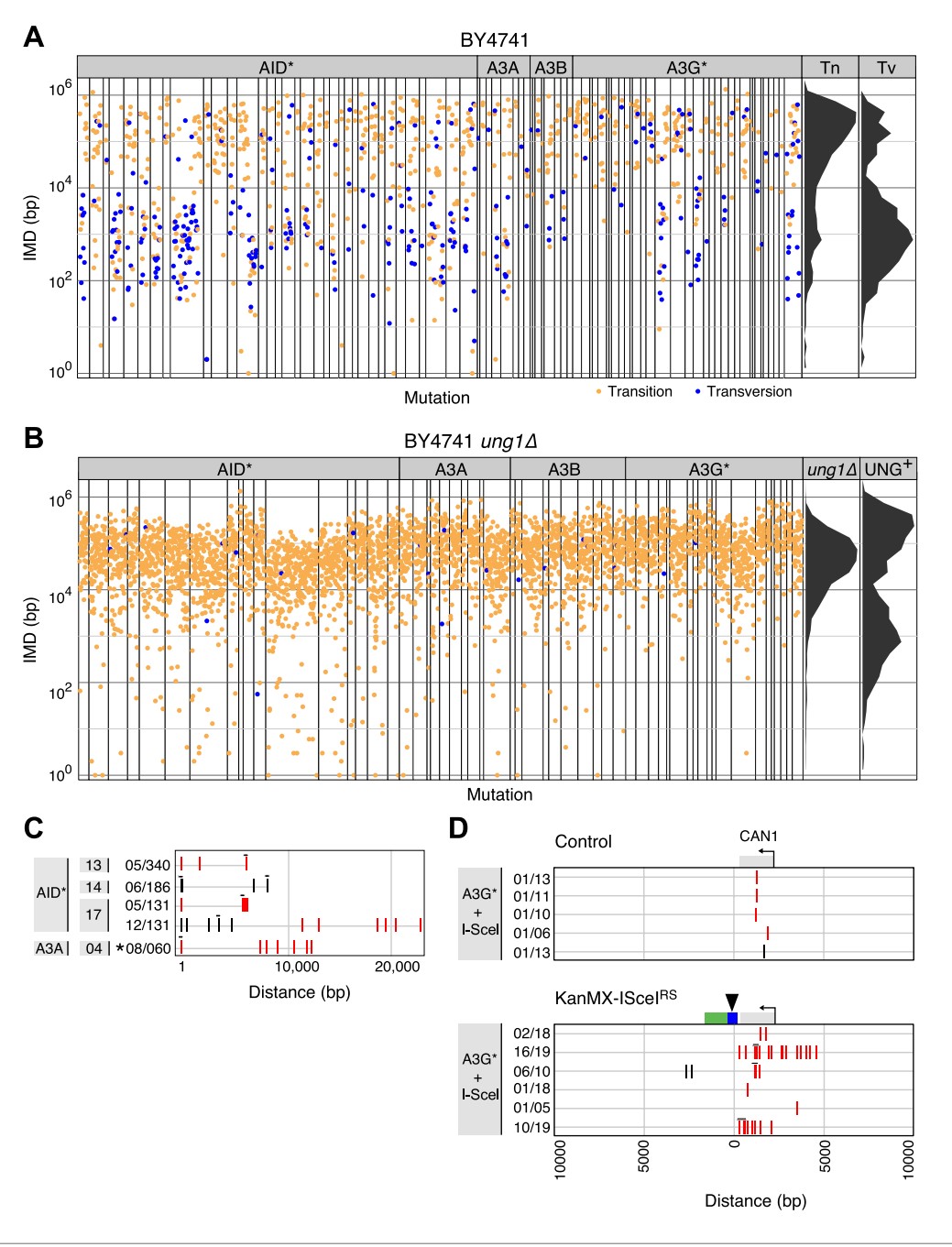

**Figure 2**. Yeast kataegic clusters are associated with transversions, are reduced by UNG-deficiency and can be triggered by a double strand DNA break. (**A**) IMD plots of AID*/APOBEC transformants reveal preferential association of nucleotide transversions with kataegic clusters. Mutation datasets and presentation are as in **Figure 1B** but with transition mutations represented by yellow dots and transversions by blue dots. Density plots depict the overall distribution of transition (Tn) and transversion (Tv) mutations at C:G pairs. (**B**) IMD plots of AID*/APOBEC-expressing *ung1Δ* yeast transformants, depicted as in **Figure 2A**. Density plots compare the distributions of IMDs in AID* transformants of *ung1Δ* and wild type yeast. (**C**) All mutation clusters identified in AID*/APOBEC3A-transformants of *ung1Δ* yeast depicted as in **Figure 1E**. (**D**) Kataegis localised to a double strand break. Mutations in the vicinity of the *CAN1* locus of (I-SceI+APOBEC3G*) transformants of either control cells or of a KanMX-ISceI[RS] derivative carrying a *CAN1*-proximal I-SceI recognition sequence. The I-SceI cut site is marked with an arrow. All *CAN1* mutations in control cells and 33/36 *CAN1* region mutations in KanMX-ISceI[RS] cells occur at the

*Figure 2. Continued on next page*

*Figure 2. Continued*

canonical APOBEC3G *CC* context. Two-thirds of the *CAN1* region mutations in the KanMX-ISceI[RS] cells were transversions.

The following figure supplements are available for figure 2:

**Figure supplement 1**. Canavanine resistance frequencies of yeast transformants carrying an I-SceI recognition sequence (I-SceI[RS]).

**Figure supplement 2**. IMD plots of mutations from Can[R] (I-SceI+APOBEC3G*) transformants of control or KanMX-ISceI[RS] cells IMD plots coloured as in **Figure 1B**.

transformants was accompanied by a dramatic shift away from mutational clustering (**Figure 2B**). Despite the fourfold 'increase' in mutation load, the percentage of mutations that are <8.5 kb from their neighbour (proximal mutations) actually 'falls' from accounting for 48% of the AID* mutations in wild type cells to 18% in *ung1*Δ transformants. Similarly, using the same criterion to distinguish clustered mutations in both datasets (≥5 linked mutations separated from their neighbour by <8.5 kb), 274 of the 1064 mutations observed in AID* wild type transformants are found within clusters compared to 28 of the 2088 mutations in the AID* *ung1*Δ transformants (**Supplementary file 1B**). Thus, the median overall IMD actually 'increases' from 13 kb in AID* wild type transformants to 41 kb in the *ung1*Δ cells despite the increase in mutation load.

These shifts do not simply reflect a fall in the proportion of clustered mutations due to the increased total mutation load. There is also an absolute fall in the amount of kataegis as judged by either the average number of clustered mutations per yeast transformant (6.9 in the wild type background vs 1.5 in the *ung1*Δ transformants) or by the frequency of kataegic events (26 kataegic stretches in 40 AID* transformants in the wild type background vs 4 kataegic stretches in 19 AID* transformants in *ung1*Δ background) (**Supplementary file 1B**). Thus, it is evident that kataegis is substantially reduced in the UNG-deficient background, but not completely lost: a few residual clusters (which exhibit evident strand polarity or bipolarity) are still detected (**Figure 2C**).

## DNA break induction stimulates APOBEC-dependent yeast kataegis

The sensitivity of kataegis to UNG-deficiency indicates that kataegis is, at least in part, triggered through the generation of abasic sites. Cleavage at abasic sites by apyrimidinic endonucleases will lead to occasional double-stranded DNA breaks: kataegis could result from AID/APOBEC deamination of single-stranded DNA exposed during the resection phase of break repair. To determine whether the DNA break repair process predispose to kataegis, we introduced a target site for the restriction endonuclease I-SceI immediately downstream of the polyadenylation site of the *CAN1* locus, and asked whether co-expression of I-SceI together with the APOBEC3G* deaminase increased the likelihood of kataegis in the vicinity. We chose to use APOBEC3G* for this experiment since it gave a good mutation load but a lower proportion of kataegic mutations than AID* (**Supplementary file 1B**): any enhancement of kataegis would therefore be more readily detectable. Consistent with previous findings (**Poltoratsky et al., 2010**), induction of I-SceI expression resulted in an increased frequency

**Table 1.** Pattern of nucleotide substitutions at C:G pairs in AID*/APOBEC yeast transformants. All mutations at C:G pairs were computed as substitutions at C

|  | AID* | | | *ung1*Δ AID* | *rev1*Δ AID* | A3A | A3B | A3G* |
|---|---|---|---|---|---|---|---|---|
|  | **Kataegic** | **Unclustered** | **Total** |  |  |  |  |  |
| C→T (%) | 46 | 87 | 76 | 99.3 | 100 | 79 | 81 | 78 |
| C→G (%) | 47 | 11 | 21 | 0.2 | 0 | 17 | 16 | 20 |
| C→A (%) | 7 | 2 | 3 | 0.5 | 0 | 4 | 3 | 2 |
| Total | 274 | 790 | 1064 | 2088 | 77 | 130 | 121 | 560 |

of deaminase-dependent selectable mutation at the linked *CAN1* locus (*Figure 2—figure supplement 1*). More importantly, in the presence of APOBEC3G*, induction of a double-strand break increases the probability that mutations in its vicinity are kataegic (*Figure 2D*).

## Comparison of yeast and breast cancer kataegis

The mutation clusters in the breast cancers were analysed in the same way as the yeast clusters. Most of the cancers identifiable by rainfall plots as harbouring major regions of kataegis also contain clusters comprising smaller numbers of same-strand nucleotide substitutions at 5′-T-C dinucleotides (*Figure 3—figure supplements 2–4*). There is some diversity amongst the breast cancers with respect to the frequency/nature of the kataegic stretches. The main outlier is tumour PD4107a which carries a dense array of highly mutated (and transition-restricted) kataegic clusters coincident with extensive genomic rearrangement in a 14 Mb region of chromosome 6 (*Nik-Zainal et al., 2012*). Overall, the kataegic clusters in the breast cancers are distributed over a similar range of lengths to those detected in the yeast transformants (*Figure 3A*) but the yeast clusters do typically contain a twofold to fivefold lower density of mutations (a mean inter-mutational distance of 1220 bp within the AID* yeast kataegic stretches compared to 209 bp in PD4107a, 335 bp in PD4103a and 763 bp in PD4199a).

## APOBEC3A and APOBEC3B deamination context preferences in yeast are similar to that in several breast cancers kataegic regions

The vast majority of the breast cancer kataegic mutations occur at C residues preceded by a T (*Figure 3—figure supplements 2–4*). In tumours PD4103a, PD4107a and PD4199a, over 91% of the kataegic C mutations are preceded by a T (*Figure 3C* and *Figure 3—figure supplement 2–4*). However, any sensitivity to the identity of the base at position −2 is exceedingly mild (average across the kataegic stretches in these three tumours is A:C:G:T = 32:20:19:29 compared to the human genome average of 30:20:20:30) (*Figure 3C*).

Previous experiments in which AID/APOBEC deaminases have been used to mutate specific bacterial or retroviral gene targets have revealed that individual deaminases show characteristic flanking nucleotide preferences. However, none of the deaminases analysed to date (AID, APOBEC1, APOBEC3C, 3DE, 3F and 3G) has been shown to exhibit a preference that accords with the breast cancer kataegic mutations. Their flanking sequence preferences (reviewed in *Conticello et al., 2007*) are either radically different (e.g., AID prefers A/G at −1; APOBEC3G prefers C at −1) or else they do not show the high (>90%) preference for T at −1 coupled to a relative indifference to the base at −2.

The mutation spectra obtained in yeast allow the consensus motifs for individual deaminases to be refined owing to the large number of potential target sequences interrogated when mutational specificity is analysed on a genome-wide basis (*Figure 3B*). With AID, APOBEC3C and APOBEC3G the yeast data essentially confirm the previously identified sensitivity to nucleotides located at positions −1 and −2 (AID: 5′-WRC, APOBEC3C: 5′-TYC, APOBEC3G: 5′-CCC) whilst allowing more precise quantitation of the degree of preference. With regard to APOBEC3A and APOBEC3B, the results reveal that (consistent with earlier studies on APOBEC3B; *Bishop et al., 2004*), both enzymes strongly prefer a T at position −1 (91%). However, the yeast studies reveal that unlike other deaminases, both APOBEC3A and APOBEC3B show mild discrimination with regard to the bases located at position −2 (APOBEC3A, A:C:G:T = 25:26:7:42; APOBEC3B, A:C:G:T = 35:14:20:31) (*Figure 3B*).

Comparing the contexts of the mutations obtained with the individual deaminases in yeast to those of the kataegic mutations in the cancers reveals that APOBEC3B has a signature that fits extremely well with the kataegis in PD4107a and PD4103a whereas APOBEC3A fits better with PD4199a (p values in all three cases <0.005) (*Figure 3D*). Interestingly, a marked bias towards a 5′-T is also seen amongst the individual singlet C mutations in several of the breast tumours (e.g., PD4199a, PD4005a and PD4120a; *Figure 3C* and *Figure 3—figure supplement 2–4*).

## APOBEC3B is highly expressed in breast cancer cell-lines and, like APOBEC3A, can cause genomic damage in mammalian cells

Although APOBEC3A has been shown to be capable of causing genomic damage in mammalian cells (*Vartanian et al., 2008*; *Stenglein et al., 2010*; *Landry et al., 2011*), the same has not been shown for APOBEC3B. We find that induction of APOBEC3B expression in stably transfected human KBM7 cells (like that of APOBEC3A) results in loss of viability as well as in genomic DNA damage as judged by the induction of γH2AX (a marker of the DNA damage response) and of 53BP1 foci (which identify DNA breaks) (*Figure 4A,B*).

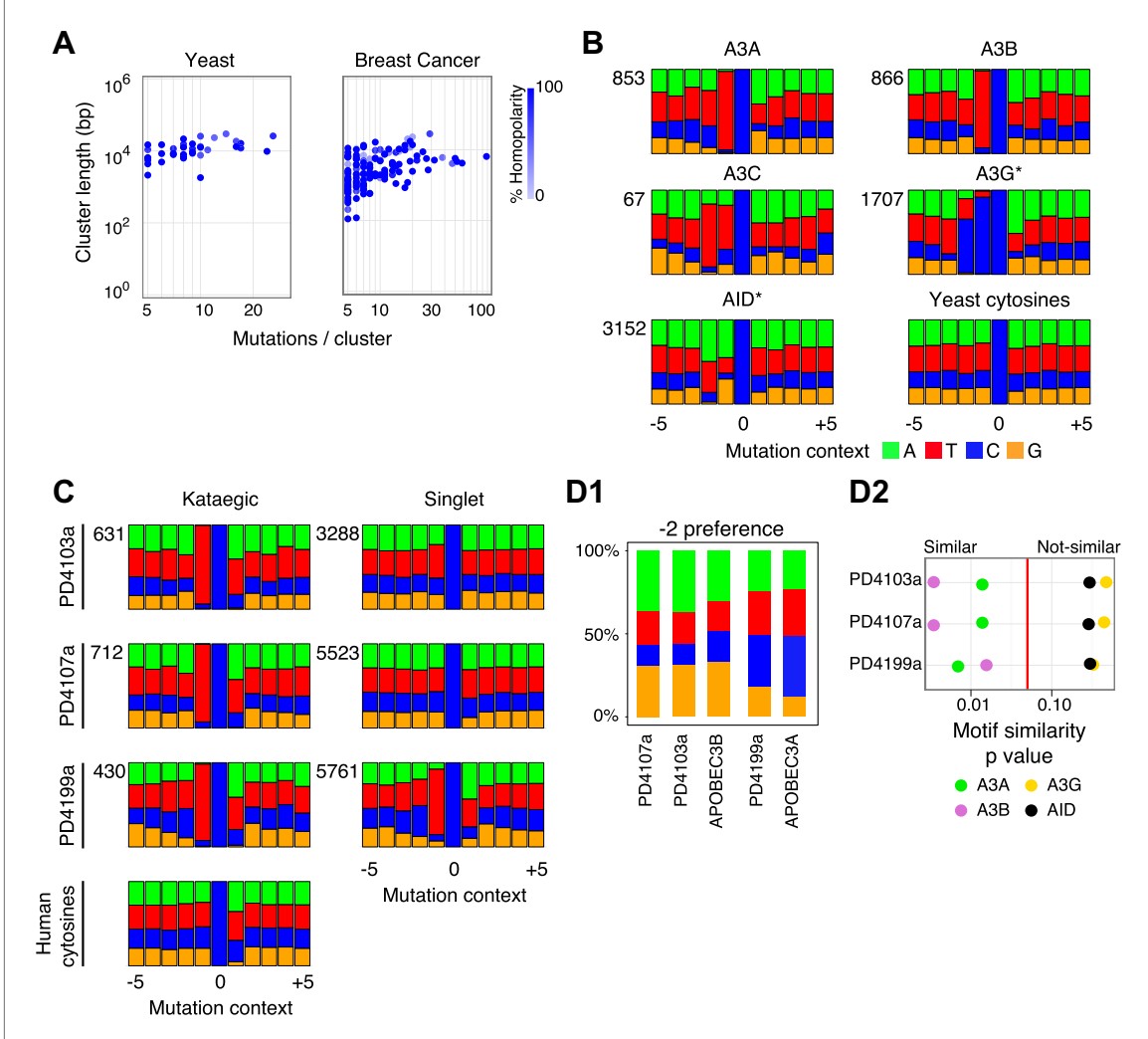

**Figure 3**. Comparison of kataegic mutations in yeast AID/APOBEC transformants with those in breast cancers. (**A**) Comparison of the length, number of mutations and polarity of yeast kataegic clusters with those in breast cancers. The degree of strand polarity is indicated by colour intensity. The breast cancer data (***Nik-Zainal et al., 2012***) are a compilation from three tumours (PD4103a, PD4107a, PD4199a) chosen for their large number of clusters. (**B**) Context of the genome wide mutated C bases in yeast AID/APOBEC transformants with total numbers of mutations in each dataset indicated. (**C**) Context of the kataegic and singlet mutated C bases in selected breast cancers. Analyses of all sequenced breast cancers are presented in ***Figure 3—figure supplements 2–4***. (**D**) Similarity of sequence contexts of C mutations in breast cancer kataegic stretches compared to those of deaminase-induced C mutations in yeast. (**D1**) Identity of the base at the −2 position of *TC* mutations in cancer kataegic regions and in APOBEC3A/B yeast transformants. The base compositions were normalised to the genomic base composition of the −2 base at *TC* dinucleotides. (**D2**) Sequence contexts similarity p-value at positions (−1 plus −2) to the mutated Cs. The contexts of all Cs throughout the yeast and human genomes are included for comparison. Mutation context of wild type versions of AID and APOBEC3G are shown in ***Figure 3—figure supplement 1***. Analysis of additional yeast transformants and breast cancers is shown in ***Figure 3—figure supplements 2–4***.

The following figure supplements are available for figure 3:

**Figure supplement 1**. Mutation context of hyperactive APOBEC3G and AID are identical to the wild type proteins.

**Figure supplement 2**. Analysis of kataegic stretches and mutation distributions of 21 breast cancers.

**Figure supplement 3**. Analysis of kataegic stretches and mutation distributions of 21 breast cancers.

**Figure supplement 4**. Analysis of kataegic stretches and mutation distributions of 21 breast cancers.

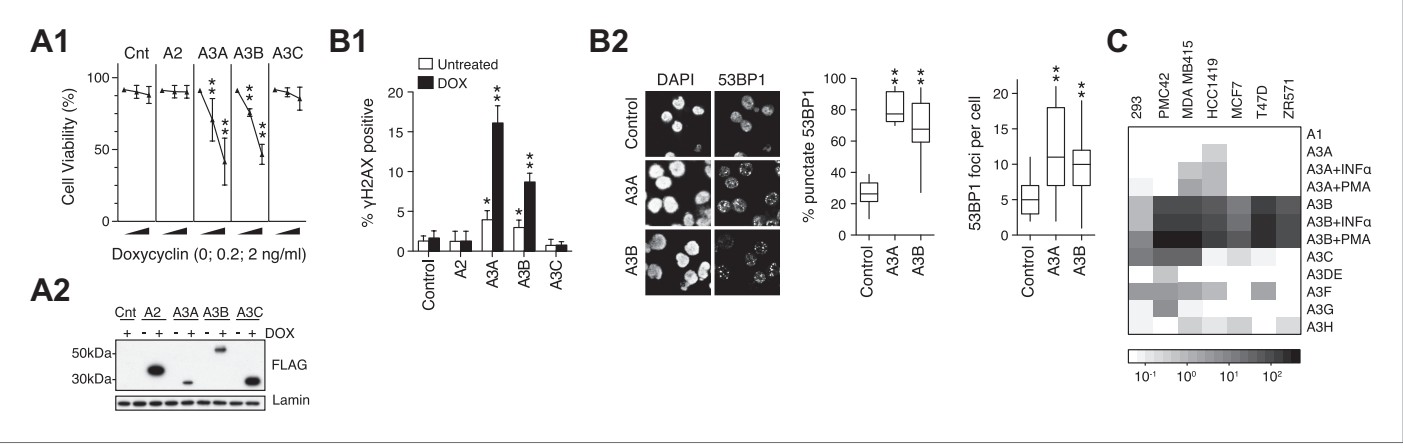

**Figure 4**. DNA damage by APOBEC3 family members and expression in breast cancer cell-lines. (**A**) Effect of enforced APOBEC expression on cell viability. (**A1**) Stable transfectants of KBM7 cells that inducibly express APOBEC proteins were incubated with inducer (doxycyclin) and viability was monitored after 72 hr. (**A2**) Expression of FLAG-tagged APOBECs after 24 hr doxycyclin treatment. (**B**) Enforced expression of APOBEC3A and APOBEC3B leads to induction of histone γH2AX and of 53BP1 foci. The percentage of cells (**B1**) positive for histone γH2AX expression quantified by flow cytometry and (**B2**) exhibiting punctate rather than diffuse 53BP1 staining quantified by immunofluorescence microscopy, with the foci number per cell indicated. (**C**) Expression of APOBEC family members in six human breast cancer cell-lines as well as in HEK293 cells was analysed by qRT-PCR of total cellular RNA. Expression is shown relative to the average of housekeeping genes HPRT and HMBS. The effect of phorbol ester (PMA) and interferon alpha (INFα) treatment on APOBEC3A and APOBEC3B levels is also shown. In all cases, * indicates p<0.1, ** indicates p<0.0001 compared to control (unpaired t-test).

Although kataegis could easily have resulted from a transient spike in deaminase expression during tumour development, it was interesting to ascertain whether APOBEC3A or APOBEC3B expression could be detected or induced in breast cancer-derived cells. RNA analysis revealed that although several APOBEC3s can be expressed in individual breast cancer cell-lines, the highest and broadest pattern of expression was evident with APOBEC3B (*Figure 4C*). Consistent with studies in other cell-types (*Madsen et al., 1999*; *Koning et al., 2009*; *Stenglein et al., 2010*), the expression of APOBEC3A and APOBEC3B in some of the breast cancer cell-lines could be enhanced by treatment with phorbol ester or interferon alpha.

## Discussion

Expression of AID/APOBECs cytidine deaminases in yeast generates mutations across the genome, a proportion of which are found in clusters. Since completing this work, two other groups have also demonstrated that cytidine deaminases can generate such clustered mutations (*Chan et al., 2012*; *Lada et al., 2012*). Here we extend on these findings, demonstrating the similarity of yeast and breast cancer kataegis, use yeast genetics to gain insight into the mechanism of kataegis and provide evidence identifying the individual APOBECs likely responsible for the kataegis in the breast cancers.

The stimulation of local kataegis in yeast by the induction of an I-SceI break indicates that the process occurs during DNA break repair, most likely by AID/APOBEC-catalysed deamination of cytidines exposed on single-stranded DNA during the resection phase of homology-mediated repair. The lengths of the kataegic stretches (mostly in the range 6–15 kb) are in the same order as the extent of resection observed during yeast DNA break repair (reviewed in *Paull, 2010*) although the occurrence and detection of kataegis is likely to bias towards longer stretches. The distances separating adjacent mutations within the yeast kataegic stretches (averaging about 1.2 kb in the AID* dataset) might in part reflect that the deaminase both jumps and slides on ssDNA, acting with possibly low efficiency at each encountered cytidine as proposed by Goodman (*Chelico et al., 2006*).

In the absence of an induced double-strand break, kataegis in AID/APOBEC-expressing yeast was greatly dependent on UNG, likely reflecting that kataegis under these circumstances was dependent on breaks generated through the processing of abasic sites. That some residual kataegis is still

observed in the absence of UNG might well reflect that breaks will occasionally occur spontaneously through other means.

The finding that a double-strand break can be the nucleating lesion for kataegis in this yeast experimental system is consistent with the close association of kataegis and rearrangements in breast cancer (*Nik-Zainal et al., 2012*). Whereas the yeast data demonstrate that double-strand breaks can nucleate kataegis, it is probable that APOBEC-catalysed kataegic deamination in exposed stretches of single-stranded DNA in the cancer cells might itself lead to DNA breaks.

It has long been known that recombinational repair of double-strand breaks in yeast is associated with an increased frequency of local mutations with implication of error-prone polymerases (*Strathern et al., 1995*). In our experiments, the signatures of the mutations associated with the I-SceI break (see *Figure 2D* legend) implicate APOBEC3 activity rather than error-prone polymerases as the source of mutations during the double-strand break repair. More recently Gordenin and colleagues have shown that extensive clusters of mutations can be induced in yeast by alkylating agents acting on single-stranded DNA (*Roberts et al., 2012*). Thus, the AID/APOBEC-mediated kataegic hypermutation, driven by these endogenous mutagens, can be viewed as a specialised, albeit dramatic, example of localised hypermutation caused by exposure of single-stranded DNA during homologous recombination, along the lines proposed by Roberts (*Roberts et al., 2012*).

It is striking that transversions in yeast are specifically associated with kataegic stretches whereas the unclustered mutations in the same cells are restricted to transitions. The reason for this is a matter for speculation but we suspect the singlet uracils largely encounter UNG as part of the base-excision repair process (which would be non-mutagenic); the C→T transition mutations would be the result of direct replication over the non-excised uracil. In contrast, the action of UNG on uracil in a stretch of exposed single-stranded DNA may yield an abasic site that is replicated over by a translesion polymerase rather than repaired.

The yeast experiments indicate that kataegis can be triggered by DNA breaks, whether generated through the joint action of the deaminase and UNG or by other processes. The same likely holds true for the breast cancer kataegis. However, there is no reason why kataegis should be restricted to such initiating triggers. One can well imagine that other processes that cause significant exposure of single-stranded DNA (e.g., DNA spooling caused by replication fork stalling [*Lopes et al., 2006*]; R-loop structures generated during transcription of suitable target sequences [*Aguilera and Gómez-González, 2008*]) could predispose to kataegis. Such mechanisms, or spontaneously-arising DNA breaks, could underlie the presence of kataegis in UNG-deficient cells (this work and *Lada et al., 2012*). A more extensive study of the genetic dependence of kataegis and of the localisation of the kataegic stretches in yeast may give insight into such possibilities.

Comparison of the yeast and breast cancer data reveals that the kataegic stretches in both sets extend over a similar range of lengths but with the cancer kataegis displaying a twofold to fivefold higher average mutation density. This could reflect differences in deaminase activity in the two organisms. It also appears that those cancers which harbour kataegic stretches comprising larger numbers of mutations additionally contain multiple clusters with smaller numbers of *T*-C mutations (*Figure 3—figure supplements 2–4*). The marked bias towards a 5′-*T* seen amongst some cancer singlet C mutations suggests that kataegis might be signalling a much wider implication of APOBEC-mediated deamination in genome-wide mutagenesis in some tumours.

The mutation data obtained in yeast reveal APOBEC3B and APOBEC3A as the only deaminases characterised whose target specificity matches the breast cancer kataegic mutations, arguing very strongly for an involvement of these deaminases in cancer kataegis. The implication of APOBEC3A fits with data from others revealing that enforced expression of APOBEC3A (as well as APOBEC3C and 3H) can lead to mutation of human papilloma viral DNA (*Vartanian et al., 2008*) as well as of transfected plasmid DNA (*Stenglein et al., 2010*). Enforced expression of APOBEC3A has also been shown to lead to genomic damage in the nucleus (*Landry et al., 2011*). The target-specificity data implicating APOBEC3B in the breast cancer mutation is not only supported by our demonstration that its enforced expression can yield DNA damage but also by the fact that it is well expressed in breast cancer cell lines. Furthermore, after submission of this manuscript, Burns et al., have demonstrated that APOBEC3B expression also correlates with a *T*-C mutator phenotype in many primary breast cancer tumours (*Burns et al., 2013*).

Thus, APOBEC3B and/or APOBEC3A are the deaminases likely responsible for the breast cancer hypermutation although it remains possible that other APOBEC3s might contribute to genome

mutation in other tumours. With regard specifically to kataegis, given that double strand breaks are a common feature of tumour development, it will obviously be interesting to discover whether whole genome sequencing of other tumour types also reveals evidence of kataegic hypermutation and whether, in light of the fact that the AID/APOBEC family has undergone considerable expansion in primates, such kataegic hypermutation might also have contributed more generally to recent genome evolution.

## Materials and methods

### Yeast transformants

Yeast strain BY4741 (*MAT*a; *his3Δ1*; *leu2Δ0*; *met15Δ0*; *ura3Δ0*) and the *ung1Δ::kanMX4* derivative were from Euroscarf (Frankfurt, Germany). The r*ev1Δ::LEU2* derivative was generated by homologous recombination to remove the open reading frame of REV1 using a LEU2 cassette generated by PCR using 157-bp 5′ homology arm and 200-bp 3′ homology arm. The *CAN1::KanMX-ISceI*$^{RS}$ strain was generated by inserting a 1.4-kb module containing the I-SceI-recognition site and the KanMX selection cassette (*Wach et al., 1994*) immediately after its poly-A site by homologous recombination. Correct integration of the targeting constructs was confirmed by PCR.

Yeast transformants expressing galactose-inducible human AID/APOBEC proteins were generated by transformation with the appropriate pRS426-derived expression vectors (*Christianson et al., 1992*) in which C-terminally-FLAG-tagged AID/APOBEC cDNAs flanked by a GAL1 promoter and tADH polyA site had been inserted between the HindIII and XhoI sites. The cDNAs encoded the full-length human wild type polypeptides except that AID* and A3G* correspond to upmutants AID-7.3 and A3G-T283I in *Wang et al., 2009*, with a FLAG-tagged A3G* comprising just the second deaminase domain used in the I-SceI experiments. For these experiments, the I-SceI-ORF with an N-terminal HA tag and 3xNLS (*Johnson et al., 1999*) was cloned between the EcoRI and XhoI sites in pSH62 (*Gueldener et al., 2002*).

For canavanine resistance assays, single yeast colonies (at least 12 independent colonies for each experiment) that had been grown overnight in glucose medium to repress expression from the GAL1 promoter were diluted 1:100 into galactose-containing medium and grown for 2 days at 30°C before serial dilutions were plated onto canavanine-selection or viability plates. Colonies were counted after 3 days growth. For I-SceI-break induction, individual colonies were grown overnight in glucose medium before dilution 1:10 into raffinose-containing medium. After 4 hr growth, galactose was added to 2% and cells were cultured for a further 2 days before serial dilution and plating as above. APOBEC3G* was used in the I-SceI experiments as it gave a good mutation load but a lower proportion of kataegic mutations than AID* (*Supplementary file 1B*). Induction of protein expression both with and without the raffinose step gave similar mutation rates.

For genome sequence determination, individual Can$^R$ colonies selected as above were subcloned by streaking out on selective plates, grown for 3 days in canavanine selection media (10 ml) and DNA prepared using Gentra Puregene Yeast/Bact. Kit (Qiagen Ltd, Manchester, UK) following manufacturers instructions.

### Sample preparation and massively parallel DNA sequencing

Short insert 500-bp library construction, flowcell preparation and cluster generation was in accordance with the Illumina no-PCR library protocol (*Kozarewa et al., 2009*). 100-bp paired-end sequencing was performed on Illumina Hiseq 2000 analysers as described in the Illumina Genome Analyzer operating manual. Short insert 2 × 100 bp paired-end reads were aligned to the reference yeast genome (SacCer_Apr2011/sacCer3) using BWA (*Li and Durbin, 2009*). An average of approximately 25-fold sequence coverage was achieved for each yeast genome.

### Mutation calling

A bespoke substitution-calling algorithm, CaVEMan (manuscript in preparation) was used for calling somatic substitutions where these were identified as alleles present in an AID/APOBEC-transformant genome but absent in the parental BY4741 genome. All high-confidence mutations included in this analysis were present in more than 0.5 variant allele fractions but were more frequently present in all reads reporting that genomic position. Post-processing filters were developed to improve the specificity of substitution calling. These filters removed false positive variants that were generated by genomic features resulting in mapping errors or systematic sequencing artefacts.

All substitutions were visually assessed using a genome browser in order to ensure a high specificity of mutation-calling.

## Cluster calling

K-cluster analysis (*Hartigan and Wong, 1979*) was used to divide intermutational distances (IMDs) into two groups, which we designated distal and proximal. An IMD which excluded 99% of the distal group was then used as a threshold for cluster calling. For all the yeast analysis, the IMD threshold was set using the combined dataset of mutations from the wild type transformants. For the breast cancer analysis, IMDs combined from all tumours were used for threshold setting (except PD4120 because of its much higher mutation load). A cluster was called when a minimum of 5 adjacent mutations were identified each with IMDs below the threshold. This 5 mutation threshold was chosen since such clusters are likely to arise with a probability of <0.001 by randomly scattered mutations.

## Sequence context similarity

Sequence contexts were compared in pairwise fashion with the Tomtom motif comparison tool using Sandelin-Wasserman similarity (MEME Suite; http://tools.genouest.org/tools/meme/cgi-bin/tomtom.cgi) and are displayed as p-values.

## Analysis of APOBEC-expressing mammalian transfectants

APOBEC expressing vectors were generated by cloning the appropriate C-terminally FLAG-tagged cDNAs into a self-inactivating retroviral plasmid. The self-inactivating retroviral plasmid was generated by cloning a pTRE-(pTRE-TIGHT; Clontech, Saint-Germain-en-Laye, France)-IRES-GFP (pMX-IG) cassette into the BglII- and 3′LTR XbaI site of pMSCVpuro. The tetracycline transactivator (TET-ON; Clontech) was cloned into a modified pMSCVpuro (Clontech) which contained an IRES-mCherry cassette at the BglII–ClaI site, to generate pTET-ON-ImC. A derivative of the KBM7 human myelocytic leukemia line that stably expressed TET-ON protein was established by retroviral infection with virus particles produced from 293 cells that had been co-transformed with pTET-ON-ImC and packaging vectors using GeneJuice (Merck KGaA, Darmstadt, Germany) according to manufacturers instructions. This KBM7[pTET-ON-ImC] cell-line was then superinfected with pMSCV/APOBEC retrovirus to yield derivatives expressing the AID/APOBEC proteins under doxycyclin-inducible control. Expression of the FLAG-tagged AID/APOBEC proteins in the KBM7 transfectants was monitored by Western blot analysis of whole cell lysates after 24 hr of doxycyclin induction using HRP-conjugated anti-FLAG antibody M2 (A8592; Sigma, Gillingham, UK), probing with anti-lamin antibody (ab16048; Abcam, Cambridge, UK) as a loading control.

Stable derivatives of KBM7 cells harbouring regulatable APOBEC proteins were induced for 72 hr with doxycyclin (inducer) and viability measured by flow cytometry by DAPI exclusion. γH2AX and 53BP1 induction and localisation was analysed by flow cytometry and confocal immunofluorescence after 24hr induction with doxycyclin; caspase inhibitor (20 µM Z-VAD-FMK; Promega, Southampton, UK) was included in the cultures for γH2AX expression analysis to maintain cell viability. For γH2AX staining, ethanol-fixed cells were stained sequentially for 1 hr with anti-γH2AX (05-636; Millipore, Watford, UK) and Alexa568-conjugated anti-mouse IgG (A-11004; Invitrogen Life Technologies Ltd, Paisley, UK) prior to resuspension in PBS containing 5 µg/ml DAPI and flow cytometry. For 53BP1 staining, cells were allowed to adhere to poly-L-lysine-coated cover slips and stained using anti-53BP1 (NB100-304; Novus Biologicals, Cambridge, UK) and Alexa 568-conjugated anti-rabbit IgG (A-11011; Invitrogen) prior to mounting with DAPI. 20-30 fields per sample were imaged with a Bio-Rad Radiance 2100 confocal microscope using a 63x oil immersion objective. Images were processed using ImageJ (default settings), and cells were scored as exhibiting either diffuse or punctate staining with punctate cells further scored for the number of foci.

## RNA analysis of breast cancer cell lines

Breast cancer cell lines were kindly provided by Dr Kerstin Meyer (Cancer Research Institute, Cambridge, United Kingdom) and RNA extracted using RNeasy Plus Mini Kit (Qiagen). cDNA was prepared using GoScript Reverse Transcription System (Promega) prior to APOBEC expression quantification by qPCR using QuantiFast SYBR Green PCR Kit using an ABI ViiA-7 system (Applied Biosystems, Paisley, UK). The primers (which were selected for specificity and equivalent amplification on APOBEC ORF templates) are given in *Supplementary file 1C*.

## Acknowledgements

This work was supported by the Medical Research Council (file reference number U105178806 and a MRC Centenary Award) and the Wellcome Trust (grant reference 098051). SN-Z is a Wellcome Trust Clinical Research Training Fellow, YLW was supported by an NIH Ruth L. Kirschstein National Research Service Award (grant number F32AI091311 from the National Institute Of Allergy And Infectious Diseases [The content of this manuscript is solely the responsibility of the authors and does not necessarily represent the official views of the National Institute Of Allergy And Infectious Diseases or the National Institutes of Health]) and PJC is a Wellcome Trust Senior Clinical Research Training Fellow (grant reference 088340 MA). We would like to acknowledge Rebecca Berrens for help establishing yeast protocols and Sarah O'Meara, Stuart McLaren and Peter Ellis as well as the Core Sequencing Facility, the IT group and many other members of the Cancer Genome Project and the Core IT team of the Wellcome Trust Sanger Institute for assistance.

## Additional information

### Funding

| Funder | Grant reference number | Author |
| --- | --- | --- |
| Medical Research Council | U105178806 | Benjamin JM Taylor, Yee Ling Wu, Cristina Rada, Michael S Neuberger |
| Wellcome Trust | 098051 | Serena Nik-Zainal, Lucy A Stebbings, Keiran Raine, Peter J Campbell, Michael R Stratton |
| NIH Ruth L. Kirschstein National Research Service Award | | Yee Ling Wu |

The funders had no role in study design, data collection and interpretation, or the decision to submit the work for publication.

### Author contributions

BJMT, SN-Z, Conception and design, Acquisition of data, Analysis and interpretation of data, Drafting or revising the article; YLW, Conception and design, Acquisition of data, Analysis and interpretation of data; LAS, KR, Acquisition of data, Analysis and interpretation of data; PJC, Analysis and interpretation of data; CR, MRS, MSN, Conception and design, Analysis and interpretation of data, Drafting or revising the article

## Additional files

### Supplementary files

- Supplementary file 1. (A) Catalogue of yeast mutations. (B) Singlet and kataegic C:G mutations in yeast transformants. Kataegic mutations in all yeast transformants, are defined as those mutations that form part of a group of at least 5 mutations in which each mutation is separated from its downstream neighbour by ≤8.5 kb. All other mutations (including those that constitute the sole mutation on an individual chromosome in a particular cell) are classed as singlet mutations. (C) Primers used for APOBEC RNA RT-PCR.

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
