## [Decision Letter]

Thank you for choosing to send your work entitled “AID/APOBECs induce DNA break-associated mutation showers: implication of APOBEC3B/A in breast cancer kataegis” for consideration at *eLife*. Your article has been favorably evaluated by a Senior editor and 2 reviewers, one of whom is a member of our Board of Reviewing Editors. The Reviewing editor has assembled the following comments to help you prepare a revised submission.

The manuscript by Taylor et al presents a concise and timely description of the role of APOBEC3A and ABOBEC3B in breast cancer kataegis. Studying the observations by Nik-Zainal et al, the authors use AID/APOBEC yeast overexpression constructs and whole genome sequencing to elucidate the mutational mechanisms induced by a set of AID/APOBEC family members.

The power of this study derives from the authors' ability to utilize yeast knockout strains to study the role of specific repair proteins within the genesis of kataegic mutations, and to obtain whole-genome sequencing data from multiple mutant yeast clones. Overall the study is very well done and we do not see any major flaws. The experiments are well laid out and the manuscript is well written.

One arising issue is the recently published study by Burns et al in *Nature* (doi:10.1038/nature11881). Although the conclusions of these two studies overlap, it is clear that the manuscript by Taylor et al offers important additional mechanistic insight into the kataegic process through the key UNG deletion experiment. The findings from this experiment expose the fundamental role of base excision within kataegis and demonstrate the utility of the yeast system the authors have created.

As you will see below, the revisions requested, while important for a final submission and publication decision, are relatively minor. Given the recent emergence of a competing manuscript of which you are aware, we ask that you make these revisions without delay.

1. In the text the authors refer to the clustered mutations as showing a “strong tendency towards strand polarity (Figure 1E and F).” We are assuming that this reference is to the observation that one tends to see either all red ticks (clustered Gs) or black ticks (clustered Cs) together in a single cluster. However, it would be good to clarify this in either the text or figure legends.

2. Was the stability of the AID and APOBEC proteins in yeast measured? Is the amount of protein produced equivalent?

3. For the canavanine resistance assays, cells were grown in glucose medium, diluted, and then grown in galactose. For the I-SceI break assays, cells are transferred from glucose medium to raffinose medium for 4 hr, and then to galactose medium. Why the difference? Also, wouldn't the induction of both the human deaminases and I-SceI be better if, instead of using glucose medium at all, cells were grown in the non-inducing/non-repressing mediums of either raffinose or 3% glycerol 3% lactate, followed by the addition of galactose?

4. The absence of REV1 could have pleiotropic effects not related to its deoxycytidyl transferase activity. For example, you may want to note previous work sequencing *CAN1* mutations in a *rev1*-null background (Mito et al., Genetics 179: 1795-1806 August 2008). In that work, a temperature sensitive mutant of polymerase delta (*pol3-t*) was shown to lead to an increase in canavanine resistance that was almost completely suppressed in the absence of REV1, without significant changes to the number of transitions or transversions. Also the absence of REV1 suppressed the increased frequencies of chromosome loss, interhomolog exchange, and direct repeat recombination measured in *pol3-t*.

5. In Figure 2D, the authors might also want to show the condition; here I-SceI is present but *not* APOBEC3G. This would add greatly to the paper because it would indicate how much of the mutation localization is due to APOBEC3G (after I-SceI).

6. The authors infer long zones of end resection, but this is only an indirect inference based on AID and Apobec action. The single-strandedness could arise from increased breathing in zones of “opened up” chromatin due to the repair at the DSB. The authors should acknowledge this additional interpretation.

---

## [Author Response]

*1) In the text the authors refer to the clustered mutations as showing a “strong tendency towards strand polarity (Figure 1E and F).” We are assuming that this reference is to the observation that one tends to see either all red ticks (clustered Gs) or black ticks (clustered Cs) together in a single cluster. However, it would be good to clarify this in either the text or figure legends*.

We agree that this is an essential point that should be heavily emphasised. We have amended the paragraph to read: “Like the cancer kataegis, the clustered mutations in the various yeast transformants showed a strong tendency towards strand polarity; mutations within a cluster occur predominantly at either a C residue or a G residue with over 88% of mutations being strand coordinated (Figure 1E and F).”

*2) Was the stability of the AID and APOBEC proteins in yeast measured? Is the amount of protein produced equivalent*?

We have measured proteins levels and there is little correlation between protein levels and the extent of deamination. We have confirmed that all AID/APPOBEC proteins are expressed in yeast. We find that AID and AID* are expressed at substantially lower levels in the yeast transformants than the APOBECs and we have included an extra panel in Figure 1–figure supplement 1 to show this.

We suspect that the number of mutations is dependent more on the specific activity of each protein rather than on its abundance, which is in keeping with observations that we have previously made using AID upmutants in mammalian cell transfectants (Wang et al., Nat Struct Mol Biol 16, 769–776 [2009]).

*3) For the canavanine resistance assays, cells were grown in glucose medium, diluted, and then grown in galactose. For the I-SceI break assays, cells are transferred from glucose medium to raffinose medium for 4 hr, and then to galactose medium. Why the difference? Also, wouldn't the induction of both the human deaminases and I-SceI be better if, instead of using glucose medium at all, cells were grown in the non-inducing/non-repressing mediums of either raffinose or 3% glycerol 3% lactate, followed by the addition of galactose*?

We were concerned that low levels of expression of AID/APOBEC proteins might result in selection for inactivating mutations on the AID/APOBEC plasmids. We therefore wanted to ensure limited and controlled exposure of the genomes to the exogenous mutators; hence the choice of glucose repressive media for culture during periods of clonal expansion. The difference in induction between experiments is historical. We have since tested both induction methods and found them to produce similar results, and we have added a comment in the Materials and methods section to reflect this: “Induction of protein expression both with and without the raffinose step gave similar mutation rates.”

However, we appreciate the reviewers' comments and we can certainly use their suggestions for growth conditions in future experiments.

*4) The absence of REV1 could have pleiotrophic effects not related to its deoxycytidyl transferase activity. For example, you may want to note previous work sequencing CAN1 mutations in a rev1-null background (Mito et al., Genetics 179: 1795-1806 August 2008.) In that work, a temperature sensitive mutant of polymerase delta (pol3-t) was shown to lead to an increase in canavanine resistance that was almost completely suppressed in the absence of REV1, without significant changes to the number of transitions or transversions. Also the absence of REV1 suppressed the increased frequencies of chromosome loss, interhomolog exchange, and direct repeat recombination measured in pol3-t*.

The reason that we suspect that the effect of REV1 deficiency reflects its deoxycytidyl transferase activity is that the transversion mutations in REV1 sufficient cells are heavily biased towards C:G substitutions. While we fully accept that we cannot exclude indirect effects of REV1 deficiency, addressing the details and mechanistic function for REV1 in kataegis is beyond the scope of this paper, and will require extensive additional work. We have modified the text to reflect the possibility of a non-catalytic role for REV1 adding in the Results section: “This presumably reflects diminished repair of the AID/APOBEC-generated uracils. There was an overall decrease in average total mutation load in AID* transformants of REV1 deficient yeast, that might reflect the possible non-catalytic roles of REV1 during DNA damage repair (Sale et al., 2012).”

*5) In Figure 2D, the authors might also want to show the condition; here I-SceI is present but not APOBEC3G. This would add greatly to the paper because it would indicate how much of the mutation localization is due to APOBEC3G (after I-SceI)*.

We agree that such a control would help emphasise the point; unfortunately we have not sequenced these genomes. The signatures of the detected mutations do, however, make this point as the vast majority occur at the APOBEC3G consensus, indicating they are caused by the activity of APOBEC3G and not other repair processes as we highlight in the figure legend. To further emphasise this, we have added the following sentence to the Discussion: “In our experiments, the signatures of the mutations associated with the I-SceI break (see Figure 2D legend) implicate APOBEC3 activity rather than error-prone polymerases as the source of mutations during the double-strand break repair.”

*6) The authors infer long zones of end resection, but this is only an indirect inference based on AID and Apobec action. The single-strandedness could arise from increased breathing in zones of “opened up” chromatin due to the repair at the DSB. The authors should acknowledge this additional interpretation*.

We do not favour an explanation based on breathing since breathing suggests a transient melting of the DNA, which would allow the unmutated DNA strand to act as a template upon strand reannealing, thereby resulting in non-mutagenic repair. Furthermore, the lengths of the observed kataegic tracts are not really compatible with measurements of DNA dynamics that suggest that fluctuations in base pairing are a relatively local phenomenon (Jose et al., Proc Natl Acad Sci USA 106, 4231–4236 [2009]). However, we fully admit that we cannot definitively exclude it; we acknowledge in the Discussion a variety of mechanisms that promote single stranded DNA could generate kataegis. In the interests of brevity, we included in the Discussion a selective but not exhaustive list of possible alternative mechanisms.